# Characterization of hepatitis B viral forms from patient plasma using velocity gradient: Evidence for an excess of capsids in fractions enriched in Dane particles

Charlotte Pronier[1], Jérémy Bomo[2], Juliette Besombes[1], Valentine Genet[2], Syria Laperche[3,4], Philippe Gripon[2], Vincent Thibault[1]*

**1** Univ Rennes, CHU Rennes, Inserm, EHESP, Irset (Institut de recherche en santé, environnement et travail) UMR_S 1085, Rennes, France, **2** Univ Rennes, Inserm, EHESP, Irset (Institut de recherche en santé, environnement et travail) UMR_S 1085, Rennes, France, **3** Department of Blood-Borne Agents, National Reference Center of Infectious Risks in Blood Transfusion, Institut National de la Transfusion Sanguine, Paris, France, **4** Etablissement Français du Sang, La Plaine-Saint-Denis, France

* vincent.thibault@univ-rennes1.fr

**Data Availability Statement:** All relevant data are within the paper and its Supporting Information files.

## Abstract

Hepatitis B virus (HBV) morphogenesis is characterized by a large over-production of subviral particles and recently described new forms in parallel of complete viral particles (VP). This study was designed to depict circulating viral forms in HBV infected patient plasmas, using velocity gradients and most sensitive viral markers. Plasmas from chronic hepatitis B (CHB) patients, HBeAg positive or negative, genotype D or E, were fractionated on velocity and equilibrium gradients with or without detergent treatment. Antigenic and molecular markers were measured in plasma and in each collected fraction. Fast Nycodenz velocity gradients revealed good reproducibility and provided additional information to standard equilibrium sucrose gradients. HBV-RNAs circulated as enveloped particles in all plasmas, except one, and at lesser concentrations than VP. Calculations based on standardized measurements and relative virion and subviral particle molecular stoichiometry allowed to refine the experimental approach. For the HBeAg-positive plasma, VP were accompanied by an overproduction of enveloped capsids, either containing HBs, likely corresponding to empty virions, or for the main part, devoid of this viral envelope protein. Similarly, in the HBeAg-negative sample, HBs enveloped capsids, likely corresponding to empty virions, were detected and the presence of enveloped capsids devoid of HBs protein was suspected but not clearly evidenced due to the presence of contaminating high-density subviral particles. While HBeAg largely influences HBcrAg measurement and accounts for two-thirds of HBcrAg reactivity in HBeAg-positive patients, it remains a 10 times more sensitive marker than HBsAg to characterize VP containing fractions. Using Nycodenz velocity gradients and standardized biomarkers, our study proposes a detailed characterization of circulating viral forms in chronically HBV infected patients. We provide evidence for an excess of capsids in fractions enriched in Dane particles, likely due to the presence of empty virions but also by capsids enveloped by an HBs free lipid layer. Identification of this new circulating viral

**Funding:** This project was supported in part by grant 16188 from ANRS (Agence Nationale de Recherche sur le Sida et les hépatites virales) attributed to Vincent Thibault and grant AIS 2018 from Rennes Metropole attributed to VT. The funders had no role in study design, data collection and analysis, decision to publish, or preparation of the manuscript.

particle sets the basis for studies around the potential role of these entities in hepatitis B pathogeny and their physiological regulation.

## Introduction

Hepatitis B virus (HBV), a DNA reverse transcribing hepatotropic virus, belongs to the *Hepadnaviridae* family [1]. Virus entry involves direct interaction between the Na+ taurocholate cotransporting polypeptide (NTCP) receptor and the large Hepatitis B surface protein (L-HBs). After intracellular trafficking through microtubules, relaxed circular DNA (RC-DNA) is released in the nucleus and converted into the so-called covalently closed circular (cccDNA), a minichromosome-like structure lifelong persisting in infected hepatocytes. cccDNA is the template for transcription of mRNAs and pregenomic RNA (pgRNA) that are then exported to the cytoplasm for translation. pgRNA is encapsidated and reverse-transcribed to generate new RC-DNA. Then, either RC-DNA returns to the nucleus for cccDNA pool replenishment or is assembled to form new virions released by exocytosis [2]. Release of virions implies multivesicular bodies and ESCRT pathway [3].

HBV morphogenesis follows several complex pathways [4]. In parallel to infectious viral particles (VP), composed of HBs-enveloped capsid-containing RC-DNA and known as 42 nm diameter Dane particles, non-infectious subviral particles (SVP), circulating as 22 nm diameter spheres or filaments, are produced in large excess. Spheres are released *via* the secretory pathway while filaments are released like virions [5]. SVP play a role as immune regulator and contribute to the persistence of infection. Current antiviral analogs have little effect on SVP production for two main reasons, first HBsAg can be translated from integrated HBV DNA and second, cccDNA decreases very slowly during therapy. In addition to VP and SVP secretion, other types of HBV particles have been recently identified *in vivo* or *in vitro*. Empty virions (EV) consisting of genome free HBs-enveloped capsids, have been found in large amounts in patient's blood and in culture models [6]. Naked empty capsid are regularly released into cell culture supernatants but it is less clear whether such viral forms circulate in plasma of infected patients [7]. Although it is accepted that RNA-containing particles also circulates in the bloodstream, uncertainties remain about the exact composition of circulating RNA species and the nature of circulating RNA-containing particles. The precise circulating HBV-RNA forms have not been completely investigated. Circulation and stoichiometry of all these particles may broadly vary according to patient, genotype and phases of infection.

The outcome of HBV infection is linked to both viral and host factors and persistent infection may lead to liver cirrhosis and hepatocellular carcinoma (HCC). Nine HBV genotypes (A-I) have been currently identified, based on more than 7.5% nucleotide divergence. HBV genotypes are heterogeneously distributed worldwide and are therefore unequally distributed among populations with heterogeneous genetic backgrounds. Consequently, a link exists between viral genotypes, natural history and treatment response [2,8]. Genotype D, itself subdivided in several subgenotypes, has a broad and overlapping geographic distribution. Genotype D chronic infection, mostly HBeAg negative, is more likely to cause severe liver disease than genotype A infection [9]. By contrast, genotype E is largely restricted to Sub Saharan Africa. Genotypes E and to a lesser extent D, are less studied compared to B and C Asian genotypes. Nevertheless, genotypes D and E are of interest, particularly in Europe as they represent a large proportion of the circulating strains. There are two major entities of chronic HBV infection according to the HBeAg serological status [10]. However, few data are available on the nature of circulating particles according to the HBeAg status or the genotype.

New biomarkers have emerged these last years for HBV management including Hepatitis B core-related antigen (HBcrAg) and HBV-RNA. Such markers should contribute to a more accurate classification of the disease, differentiating at-risk patients to those with more favorable outcomes and leading to individualized care of patients. HBcrAg is a composite marker composed of HBcAg, HBeAg and a 22 kDa core-related protein (p22cr) [11]. Higher HBcrAg levels are measured in HBeAg positive patients than in HBeAg negative. HBV-RNAs are detected in the serum of patients at different levels [12]. Despite heterogeneous quantification protocols, the existence of circulating HBV-RNA is now admitted. These markers have recently been studied as tools to classify patient disease stage status or progression to HCC, to predict HBeAg and HBsAg loss or reactivation during immunosuppression and to evaluate treatment efficacy or chance of HBsAg clearance after treatment discontinuation [13]. The objective of the study was to depict circulating HBV viral forms in patients' plasma, using a new efficient and rapid system that combines use of velocity gradients and recently developed markers, HBcrAg and HBV-RNA.

## Results

Four plasmas from four blood donors with genotype D (n = 2) or E (n = 2) HBV infection were selected. For each genotype, one plasma was HBeAg positive and the other negative. As shown in Table 1, all plasmas were fully characterized for all available markers. Notably, plasma B7686 (HBeAg negative, genotype D) had the lowest values for all measured parameters in particular for RNA. For all plasmas, HBV-DNA levels were approximately 3.7 log above the HBV-RNA viral load. VP$_{DNA}$, the calculated amounts of VP according to DNA measurement and SVP$_{HBs}$, the calculated amounts of SVP according to HBsAg concentration were determined. Calculated ratios of SVP$_{HBs}$ on VP$_{DNA}$ for each plasma showed an overproduction of SVP thousand to hundred thousand folds over VP.

Nycodenz velocity gradients were first performed on each of these plasmas. Then, on each collected fraction, HBV-DNA, HBV-RNA, HBsAg and HBcrAg were determined. The amount of each marker in one fraction is graphically represented as the percentage of the total amount collected from all fractions (Fig 1). Within the velocity gradient, density values ranged from 1.264 to 1.027 g.cm$^{-3}$ and gradient ranges were highly reproducible between experiments (S1 Fig). A DNA peak was reproducibly identified in fractions 4 to 6, likely corresponding to concentrated Dane particles (VP), for all tested samples. The single HBsAg peak observed in fractions 11 to 13, for all samples, likely identified SVP. Noteworthy, the richest DNA fraction for each plasma was associated with slightly detectable HBsAg, with values below 17 IU/mL. By contrast, a first HBcrAg peak with HBcrAg levels above 6 log U/ml, was detected in VP-enriched fractions (lower fractions; F4-F6). This detection was observed for all tested plasmas,

**Table 1. Virological characteristics of the 4 selected plasmas.**

| Plasma | Genotype (HBeAg status) | HBV DNA (log IU/mL) | HBV RNA (log U/mL) | HBsAg (log IU/mL) | HBcrAg (log U/mL) | VP$_{DNA}$ (number/mL) | SVP$_{HBs}$ (number/mL) | SVP$_{HBs}$/VP$_{DNA}$ |
|---|---|---|---|---|---|---|---|---|
| B7195 | D (+) | 9.31 | 5.95 | 4.25 | 7.9 | 1.02E+10 | 1.43E+13 | 1 401 |
| B7505 | E (+) | 9.40 | 5.37 | 4.49 | 8.2 | 1.26E+10 | 2.48E+13 | 1 978 |
| B7207 | E (-) | 8.21 | 5.21 | 4.00 | 7.2 | 8.11E+08 | 8.04E+12 | 9 915 |
| B7686 | D (-) | 6.74 | 1.89 | 3.73 | 5.5 | 2.75E+07 | 4.32E+12 | 157 140 |

VP$_{DNA}$ corresponded to the calculated amount of VP according to DNA measurement. SVP$_{HBs}$ corresponded to the calculated amount of SVP according to HBsAg concentration.

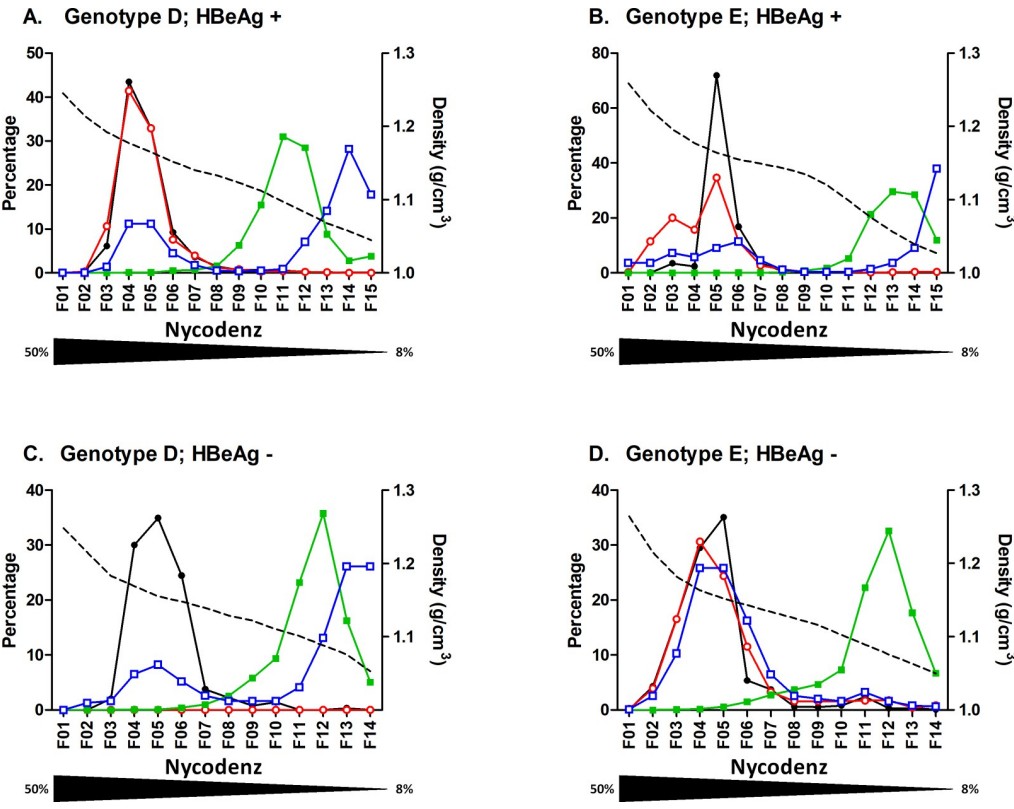

**Fig 1. Nycodenz velocity gradient profiles of 4 HBV-positive plasmas.** A. genotype D HBeAg positive plasma; B. genotype E HBeAg positive plasma; C. genotype D HBeAg negative plasma; D. genotype E HBeAg negative plasma. Density of each fraction is shown as a dashed line. Each measured markers, HBV-DNA (black circle), HBV-RNA (empty red circle), HBsAg (green square) and HBcrAg (empty blue square), are represented as percentage per fraction. Nycodenz concentration gradient is indicated by the underneath triangle.

except sample B7686 (Gt. D, HBe-) with a crude circulating HBcrAg value of 5.5 log U/mL. For this sample, fractions enriched in VP contained up to 3.8 log U/ml HBcrAg, slightly less than for the other plasmas. For both HBeAg positive samples, a second HBcrAg peak was observed in the upper fractions, above the HBsAg peak (fractions 14–15), likely corresponding to HBeAg as confirmed by a specific HBeAg assay (S1 Table in S1 File). Surprisingly, on HBeAg negative samples (B7207 and B7686), HBcrAg was also detected and quantified in the last upper two fractions (F13-14) with concentrations around 4.4 log U/ml in each of these fractions (S4 & S5 Tables in S1 File). These values were at least one hundredfold lower than values obtained for the same fractions on HBeAg positive samples. Yet, all fractions collected from HBeAg negative plasmas were non-reactive for HBeAg when tested with the HBeAg specific assay (S1 Table in S1 File). Interestingly, HBV-RNA, DNA and HBcrAg highest values were detected in the same fractions whatever the tested plasma except for plasma B7686, characterized by a very low RNA viral load (1.89 log U/mL). Absence of RNA detection in gradient fractions obtained from this sample was attributed to values below our limit of quantification. These data suggest that fractions enriched in VP also contain RNA-containing particles but to a lesser extent.

Based on the described composition of both VP and SVP in viral components, *i.e.* HBsAg, DNA and core Ag, and their respective stoichiometry for both viral forms, the expected number of VP in the richest DNA fraction for each plasma (fraction 4 or 5 depending on the gradient) was calculated according to each measured marker (cf. material and methods). $VP_{DNA}$

**Table 2. Estimated number of VP, calculated according to DNA (VPDNA), HBc (VPHBc) or HBs (VPHBs) concentrations, and their relative respective ratios in the richest DNA fraction of velocity gradients for each plasma.**

| plasma (genotype; HBe status) | Selected DNA containing fraction # | $VP_{DNA}$ Number/ fraction according to [DNA] | $VP_{HBc}$ Number/ fraction according to [HBc] | $VP_{HBs}$ Number/ fraction according to [HBs] | $VP_{HBs}/VP_{DNA}$ | $VP_{HBc}/VP_{DNA}$ | $VP_{HBc}/VP_{HBs}$ |
|---|---|---|---|---|---|---|---|
| B7195 (Gt D; positive) | 4 | 1.47E+08 | 8.96E+08 | 2.18E+08 | 1.5 | **6.1** | 4.11 |
| B7505 (Gt E; positive) | 5 | 8.65E+07 | 1.13E+09 | 1.28E+08 | 1.5 | **13.0** | 8.82 |
| B7686 (Gt D; negative) | 5 | 5.85E+04 | 1.79E+06 | 1.04E+08 | 1777.0 | **30.6** | 0.02 |
| B7207 (Gt E; negative) | 5 | 8.65E+06 | 3.57E+08 | 7.70E+08 | 89.1 | **41.2** | 0.46 |

corresponded to the calculated amount of VP according to DNA measurement; $VP_{HBc}$ and $VP_{HBs}$ corresponded to the calculation according to HBcrAg and HBsAg concentrations, respectively. Respective ratios from these calculated values are presented in Table 2.

$VP_{HBs}/VP_{DNA}$ ratios were heterogeneous ranging from 1.5 to 1777. For HBeAg positive plasmas, $VP_{HBs}/VP_{DNA}$ ratios were close to 1, meaning that measured HBsAg likely corresponded to VP-associated HBs and attested an acceptable separation by velocity gradient between VP and SVP in each selected fraction. For HBe negative plasmas, $VP_{HBs}/VP_{DNA}$ ratios were reproducibly well above 1 (89 and 1777) despite similar gradient profiles, possibly indicating some residual HBsAg containing particles co-sedimenting with VP.

Considering the $VP_{HBc}/VP_{DNA}$ calculated ratios, they were consistently above 1 (6.1 to 41.2), suggesting that genome-free capsids were also present in these fractions. Values seemed slightly higher for HBe-negative vs. HBe-positive plasmas.

Finally, the $VP_{HBc}/VP_{HBs}$ were distributed from 0.02 to 8.82. Contrary to what observed for the other ratios, the $VP_{HBc}/VP_{HBs}$ ratios were almost 10 times higher for HBeAg-positive than negative plasmas. Analysis of these ratios supports an overproduction of genome-free capsids. Thus, we hypothesized that in HBeAg positive plasmas, VP are also accompanied by an overproduction of enveloped capsids not containing HBs.

To further characterize the RNA- and the capsid-containing particles, similar velocity gradient separation was done after a detergent pretreatment (Fig 2; S6 & S7 Tables in S1 File). Upon NP40 treatment, DNA, RNA and HBcrAg peaks shifted toward bottom fractions (fraction 1), while the HBsAg peak remained in fraction 12 and 13. This observation suggested that similarly to DNA-containing capsids (VP), RNA-containing particles and genome-free capsids were enveloped whatever the HBeAg status. Altogether, for all plasmas, results from NP-40 treatment confirmed the enveloped nature of the capsid excess.

As velocity gradients separate particles according to several parameters such as size and density, a set of experiments was undertaken using a classical equilibrium sucrose density gradient. Profiles obtained through equilibrium gradient looked similar to the ones obtained using velocity for both tested plasmas, one HBeAg positive (Gt D; HBeAg +; B7195), one negative (Gt E; HBeAg -; B7207) (Fig 3, S8 & S9 Tables in S1 File). DNA, RNA and first HBcrAg peaks were all found in fraction 5 and well separated from the HBsAg peak in fraction 11. A second HBcrAg peak in the late fractions was also characterized for both plasmas. The faint peak does not clearly appear for the second plasma on the graphic due to presentation as percentages but the dosage is positive (refer to S9 Table in S1 File).

As performed for velocity gradient, for 2 genotype E plasmas (B7505 and B7207), sucrose separation at equilibrium was done after pretreatment with a detergent. Upon NP40 treatment, a shift toward higher density fractions (fractions 1–2) was observed for DNA, RNA and

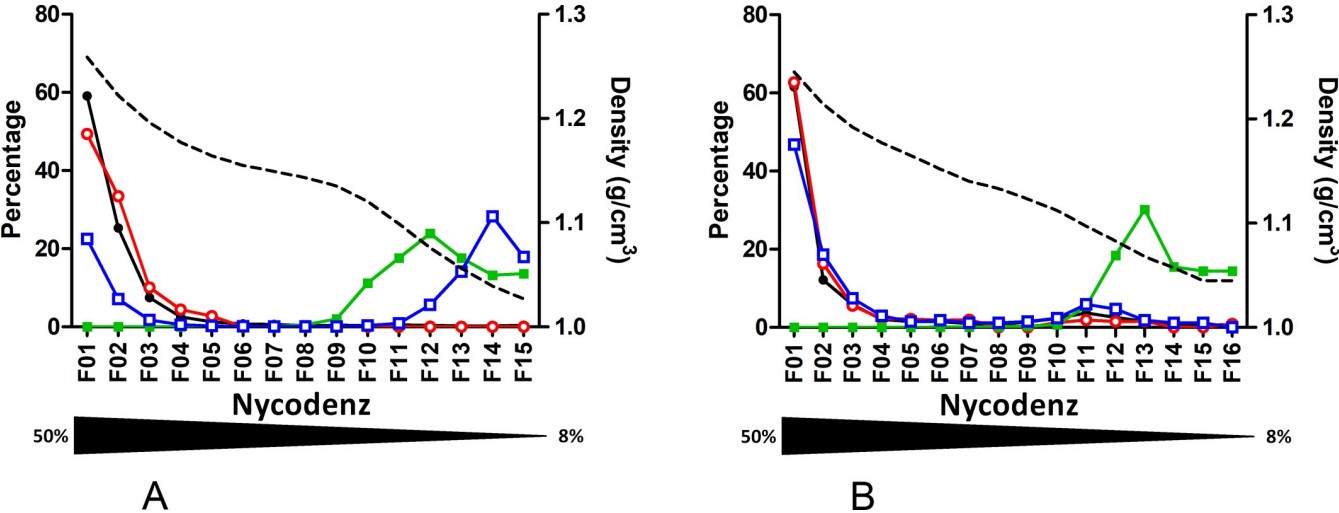

**Fig 2. Effect of NP40 treatment before fractionation.** Quantification of HBV-DNA, HBV-RNA, HBsAg and HBcrAg in Nycodenz velocity gradient fractions of plasmas treated with NP-40 before fractionation. **A.** From genotype D HBeAg positive plasma. **B.** From genotype E HBeAg negative plasma. Density of each fraction is shown as a dashed line. Each measured markers, HBV-DNA (black circle), HBV-RNA (empty red circle), HBsAg (green square) and HBcrAg (empty blue square), are represented as percentage per fraction. The Nycodenz gradient range is indicated by the underneath triangle.

HBcrAg while the HBsAg peak remained around fraction 13 (Fig 4; S10 & S11 Tables in S1 File).

To further describe the composition of VP enriched fractions for Gt E plasmas (fraction 5 from Fig 1B & 1D) sucrose density gradients (30–60% sucrose) were performed on these fractions (Fig 5; S12 & S13 Tables in S1 File). For the HBeAg positive plasma, a single peak (peaking on fraction 5) of concomitant HBsAg, DNA, RNA and HBcrAg was observed. For the HBeAg negative plasma, RNA, DNA and HBcrAg sedimented concomitantly (around fraction #7) while the peak of HBsAg was identified in lighter fractions (#9; 1.202 g/cm$^3$). This specific

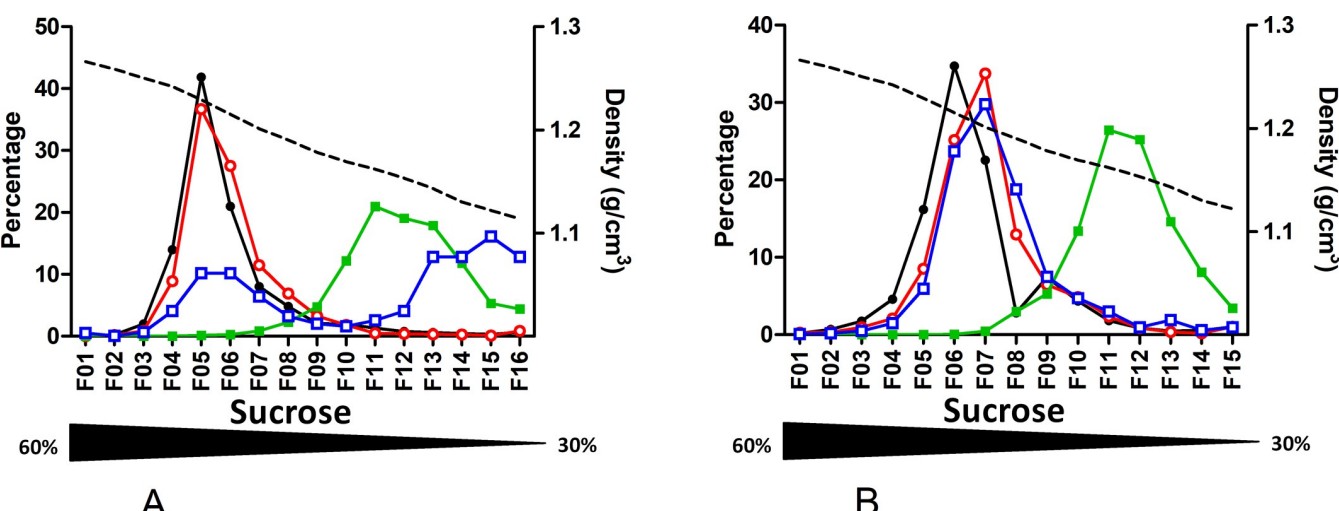

**Fig 3. Sucrose density gradient profiles of 2 HBV-positive plasmas.** Quantification of HBV-DNA, HBV-RNA, HBsAg and HBcrAg in sucrose density gradient fractions. A. genotype D HBeAg positive plasma. B. genotype E HBeAg negative plasma. Density of each fraction is shown as a dashed line. Each measured markers, HBV-DNA (black circle), HBV-RNA (empty red circle), HBsAg (green square) and HBcrAg (empty blue square), are represented as percentage per fraction. The sucrose gradient range is indicated by the underneath triangle.

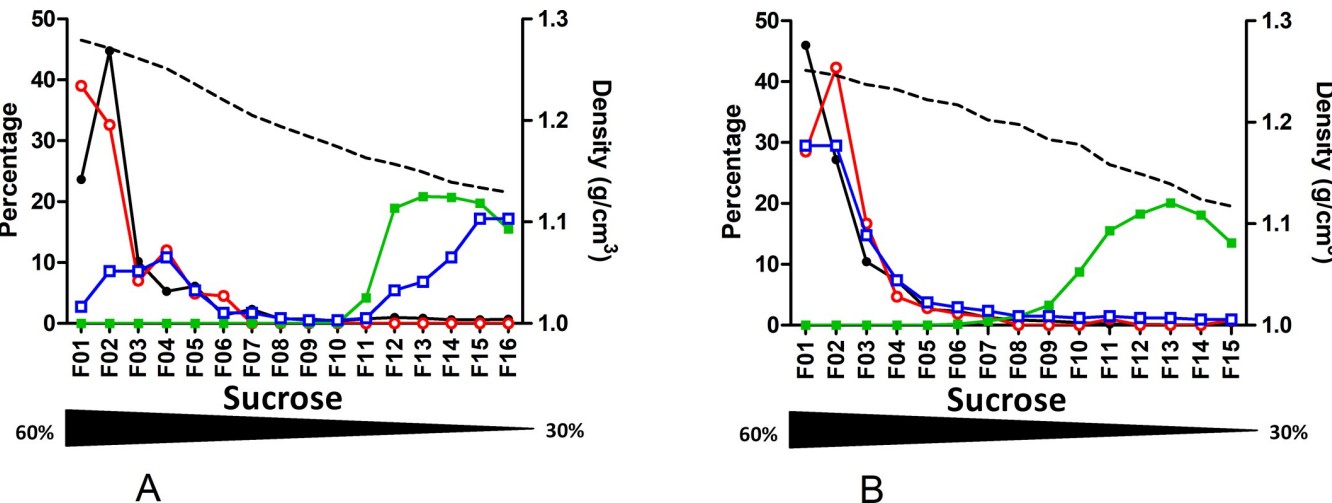

**Fig 4. Effect of NP40 treatment before fractionation.** Quantification of HBV-DNA, HBV-RNA, HBsAg and HBcrAg in sucrose density gradient fractions of plasmas treated with NP-40 before fractionation. **A.** From genotype E HBeAg positive plasma. **B.** From genotype E HBeAg negative plasma. Density of each fraction is shown as a dashed line. Each measured markers, HBV-DNA (black circle), HBV-RNA (empty red circle), HBsAg (green square) and HBcrAg (empty blue square), are represented as percentage per fraction. The Nycodenz gradient range is indicated by the underneath black triangle.

profile tends to indicate that this unique HBsAg peak relates to the residual presence of SVP despite the previous velocity separation step. According to calculations, in HBeAg+ sample, the amount of HBc containing particles is 34.6 fold higher than DNA containing particles, while HBsAg containing particles outreach by only 4.8 fold, DNA containing particles (Fig 6; Table 3). For HBeAg- sample, these figures are similar with an amount of HBc containing particles 33.3 fold higher than DNA containing particles, while HBsAg containing particles excess is more difficult to appreciate if a potential residual SVP contamination is considered. The difference between HBc containing particles and HBs containing particles, could be due to particles composed of genome-free capsids enveloped by a lipid layer not containing HBsAg.

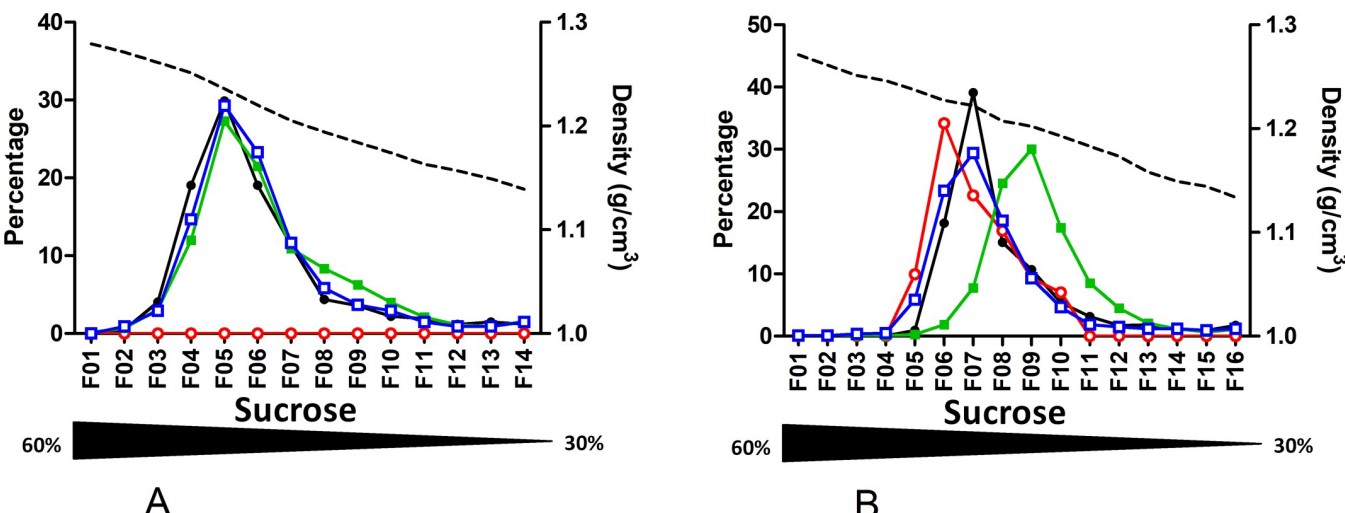

**Fig 5. Sucrose density gradient profiles of the most DNA enriched fraction from Nycodenz velocity gradients of 2 plasmas.** A. genotype E HBeAg positive plasma; B. genotype E HBeAg negative plasma; Density of each fraction is shown as a dashed line. Each measured markers, HBV-DNA (black circle), HBV-RNA (empty red circle), HBsAg (green square) and HBcrAg (empty blue square), are represented as percentage per fraction. The sucrose concentration gradient is indicated by the underneath black triangle.

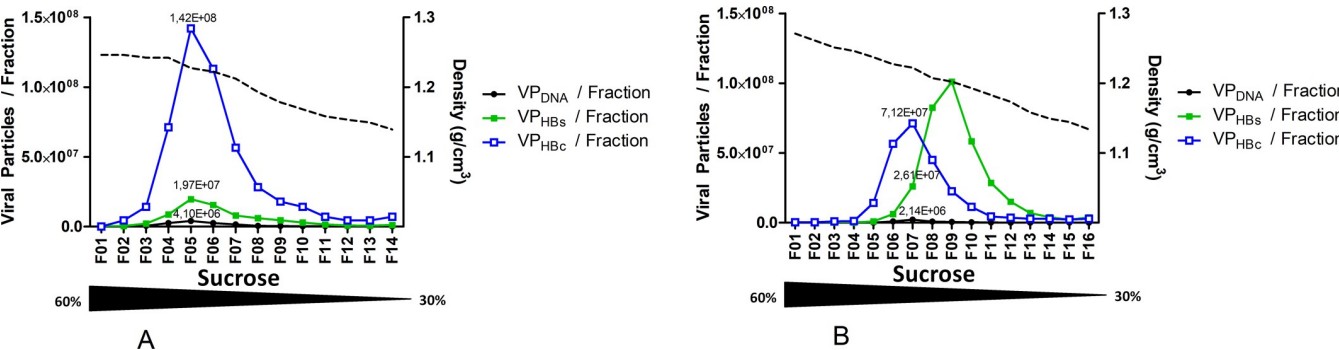

**Fig 6.** Number of Virions particles (VP), for genotype E plasmas, calculated according to DNA (VP$_{DNA}$; black circle), HBc (VP$_{HBc}$; empty blue square) or HBs (VP$_{HBs}$; green square) concentrations per fraction in a 30–60% sucrose density gradient profiles of the richest DNA fraction obtained from upstream velocity gradients.

As HBcrAg value represented the sum of several protein concentrations, we took into consideration the fractions containing mostly capsids (lower fractions) and those containing most likely HBeAg (upper fractions) to define the reactivity that could be attributed to HBc or HBe in crude plasma. A ratio of 2 for both HBeAg positive plasmas was obtained, meaning that HBeAg accounted for two-thirds of HBcrAg reactivity while HBc and/or p22cr represented the last third.

## Discussion

Nycodenz velocity gradients offer an attractive and alternative approach to study viral form distribution in patient's plasma, providing complementary information from those obtained by more conventional equilibrium sucrose density gradients. This approach allows also a faster separation. We provide original results obtained both on HBeAg positive and negative plasma from chronically infected patients with less studied genotype D and E HBV infection. To avoid any methodological bias, we started from large volumes of fully characterized plasmas from 4 blood donors, obtained from INTS, letting us the opportunity to confirm our results on several sets of experiments performed on the same raw material. Obtained gradient profiles on each plasma were highly reproducible. The original collection method through tube bottom piercing associated to a free flow, considerably reduces fluidic turbulence and potentially lowers contamination by SVP present in large excess in the upper fractions. Whatever the HBeAg status or the genotype, all velocity gradients presented a similar rate-zonal separation profile characterized by DNA-enriched bottom fractions surmounted by HBsAg-enriched fractions [14]. HBcrAg and HBV-RNA quantifications in each fraction refined previously obtained results on such gradients.

**Table 3. Estimated number of VP, for genotype E plasma, calculated according to DNA (VPDNA), HBc (VPHBc) or HBs (VPHBs) concentrations, and their relative respective ratios in the richest DNA fraction obtained from velocity gradients and further separated in a 30–60% sucrose density gradient.**

| plasma (genotype; HBe status) | Selected DNA containing fraction # | VP$_{DNA}$ Number/ fraction according to [DNA] | VP$_{HBc}$ Number/ fraction according to [HBc] | VP$_{HBs}$ Number/ fraction according to [HBs] | VP$_{HBs}$/ VP$_{DNA}$ | VP$_{HBc}$/ VP$_{DNA}$ | VP$_{HBc}$/ VP$_{HBs}$ |
|---|---|---|---|---|---|---|---|
| **B7505** (Gt E; positive) | 5 | 4.10E+06 | 1.42E+08 | 1.97E+07 | 4.8 | 34.6 | 7.2 |
| **B7207** (Gt E; negative) | 7 | 2.14E+06 | 7.12E+07 | 2.61E+07 | 12.2 | 33.3 | 2.7 |

First, we confirmed the circulation of enveloped HBV-RNA containing particles in plasmas [15]. These particles were present in lower fractions together with Dane particles, both in velocity and density gradients. However, these RNA containing particles are thousand to ten thousand less abundant than Dane particles. The lower abundance of HBV-RNA in presence of high HBV-DNA load is frequently described in the literature [16,17]. Generated profiles in sucrose density gradient are similar to those presented by Rokuhara et al. for genotype C [18]. In their work, co-localization of DNA, RNA and HBc peaks suggested that HBV-RNA was integrated into core containing particles. Unfortunately, lack of information regarding each fraction density, their collection method and the HBeAg status of the tested samples precluded any accurate comparison with our findings. More recently, Prakash et al. presented Nycodenz comparative gradient profiles with and without detergent treatment [19]. Despite a different gradient method approach and the chosen graphical representation, their profile and ours indicate a good separation between VP and SVP and the distribution of viral RNA in the same fractions as DNA, suggesting that RNA and DNA are contained in particles of similar density. Determination of HBcrAg and HBeAg in each fraction allowed us to refine this model. For all gradient profiles with an RNA peak, both HBcrAg and RNA peaks from the lower fractions co-sedimented, likely indicating that RNA is associated with capsid proteins. Detergent treatment deeply modified our gradient profiles contrary to what was reported by Prakash et al [19]. As expected for enveloped viral forms, detergent treatment led to significant sedimentation of DNA, RNA and HBcrAg markers to the bottom fractions (Figs 2 & 4). By contrast, their experiments solely showed the broadening of HBV DNA and RNA peaks, with a tendency toward higher Nycodenz concentrations migration. This discrepancy could possibly be explained by their different experimental conditions in terms of temperature, speed, time of centrifugation and type of detergent. In any case, our experiments performed with NP-40 provide further evidence that HBV-RNA is present in enveloped particles in plasmas.

If circulating HBV-RNA in some chronically infected patients is now well established, some doubts remain about the precise composition of detected viral RNA specie(s). Our study was not designed to characterize circulating RNA species. In a recent study using HBV full-length 5'RACE approach, the authors report that in CHB patients, circulating HBV-RNA seems to belong to one of 3 main categories, pgRNA, spliced pgRNA variants and HBx [20]. With the RNA detection method used in our study, adapted from Van Bommel *et al.*, all polyadenylated RNA forms including pgRNA are detected without distinction, except truncated RNAs, as we solely used the 3' long primer. One of the strengths of this study is the use of a specific HBV-RNA RT-PCR assay with all the necessary controls to attest to this. Indeed, artefactual amplification of single-stranded or double-stranded DNA is likely to occur in the situation of high viral loads. Although being a promising novel biomarker, HBV-RNA molecular characterization and assay standardization are needed to improve our knowledge and compare studies between each other.

HBcrAg is a composite marker still in quest of a consensual clinical signification [21]. Yet, there are only sparse data identifying the circulating proteins detected through this test. Only two studies based on sucrose gradient separation of an HBeAg-positive, Asian genotype, plasma and carried out by those who developed the HBcrAg assay are available [18,22]. Our study completes these data not only on HBeAg positive and negative plasmas but also on genotypes D and E. We confirm that the HBcrAg assay detects with a good sensitivity both HBc and HBe proteins. Based on virion protein composition stoichiometry and signals obtained after gradient separation, we could determine that the current HBcrAg assay (with a sensitivity of 3 logU/ml as recommended by the manufacturer) would theoretically detects around 1 million VP while an HBsAg assay with a sensitivity of 0.05 IU/mL would give a positive signal for around $9.10^6$ VP. In fractions enriched in VP, HBcrAg was detected at high concentrations

(3.8 to 6.6 log U/ml) contrary to HBsAg (2.3 to 17 IU/ml). By its design, the HBcrAg assay detects and quantifies in an equivalent manner HBc, HBe and p22cr with a better sensitivity than what can be expected from a more conventional western blot approach. With a limit of quantification of 1 kU corresponding to 10 pg of one of these 3 proteins, one may estimate that the HBcrAg assay is probably around 5000 times more sensitive than a Western blot, with an accepted limit of detection of approximately 50 ng per protein [6,11]. Ideally, it would even be more interesting to have a specific HBcAg assay.

For the HBeAg positive plasmas, two HBcrAg peaks were detected. The first one, co-localizes with nucleic acids. As NP40 treatment led to significant sedimentation of HBcrAg to the bottom fraction, both in nycodenz velocity and sucrose density gradients, it is unlikely to be HBeAg or preC proteins of high density but rather HBc detected in fractions containing enveloped particles and dissociated by detergent pretreatment (Figs 2 & 4). The second HBcrAg peak is above the major HBsAg peak (SVP) and was later confirmed as HBeAg using a specific HBeAg assay. From HBcrAg values obtained on each gradient performed on the different plasmas, one may estimate that HBeAg detection contributed for two third of the HBcrAg value, while one-third only could be attributed to HBc and p22cr reactivity. These values are in agreement with a recent study indicating that HBcrAg includes HBcAg (10%), p22cr (10%), but predominantly HBeAg (80%) [23]. Such finding should warn us about the interpretation of this marker when monitoring HBeAg positive patients. Indeed, HBcrAg concentrations most likely reflect circulating HBeAg rather than core associated proteins.

Surprisingly, for both HBeAg negative plasmas, a similar detection profile was observed with an unexpected HBcrAg signal localizing also in the upper fractions. Specific HBeAg testing of these fractions was negative as expected from HBeAg results obtained on crude plasmas. One hypothesis could be the presence of HBeAg-HBeAb complexes not detected by the HBeAg assay but identified by the HBcrAg assay. Reactivity of this latter assay would be explained by not only specific Ab targeting different epitopes but also a pretreatment step, using SDS detergent, likely disrupting Ag-Ab complexes. A second hypothesis would be the presence of preC gene products in the upper fractions, not reactive with the HBeAg assay but detected with the HBcrAg assay. Indeed, this test contains antibodies directed against most preC peptides and such peptides have been described as heterogeneous in size and density [23]. Noteworthy, HBV sequencing of plasma B7207 (HBeAg negative) revealed a precore variant (G1896A mutation) resulting in an absence of HBeAg synthesis. However, we cannot formally exclude the circulation of both wild-type and mutated strains or a recent change in HBeAg status to explain HBcrAg detection in the upper fractions in HBeAg negative samples.

Using standardized quantification methods on different fractions, we could calculate the proportion of each viral form based on their respective stoichiometric molecular composition. Analysis of ratios in VP-enriched fractions obtained on two successive gradient separation steps revealed an overproduction of genome-free capsids. Indeed, calculated amount of VP according to DNA ($VP_{DNA}$) is lower than calculated amount of VP according to HBcrAg concentrations ($VP_{HBc}$). Thus, in VP-enriched fractions of HBeAg positive or negative plasmas, there are other forms containing capsids than the sole VP (Fig 7).

For HBeAg positive plasmas, we first demonstrated that HBc proteins found in the bottom peak were enveloped (NP40 detergent experiment), then that the little amount of HBsAg found in this peak was not in agreement with all capsids being HBs-enveloped ($VP_{HBc}/VP_{HBs}$ = 7.2, or more than 1 for B7505 plasma for example). We thus raised the hypothesis that HBcrAg, and likely capsids, were enveloped within a lipid structure not containing HBsAg. Such new viral forms were tentatively named "budding capsid". Budding capsids would thus constitute an additional type of particles produced during HBV morphogenesis in HBeAg positive patients and would thus be different from previously described empty virions. The

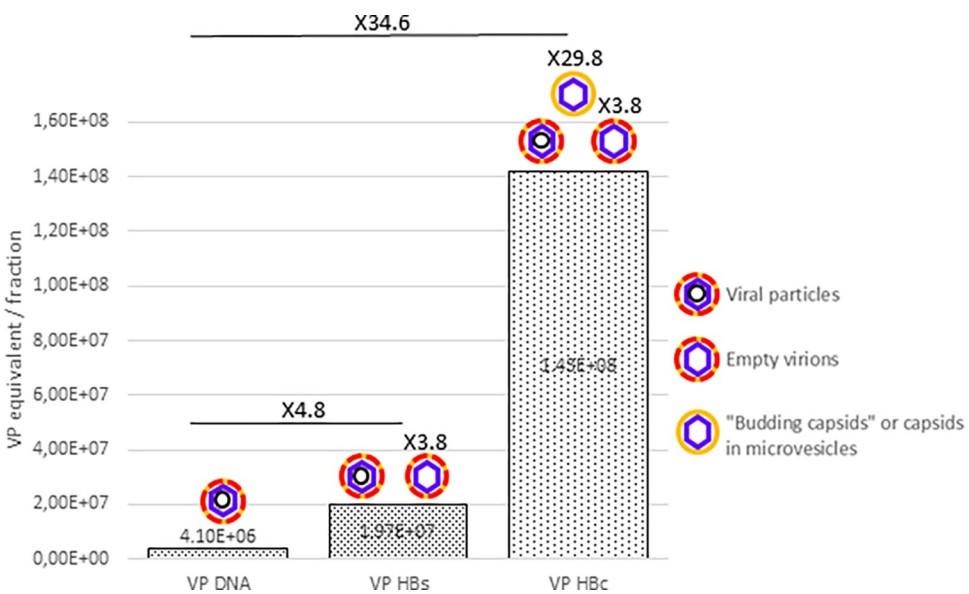

**Fig 7. Number of Virions particles (VP), for genotype E HBeAg + plasma, calculated according to DNA (VP$_{DNA}$), HBc (VP$_{HBc}$) or HBs (VP$_{HBs}$) concentrations in fraction 5 of a 30–60% sucrose density gradient profiles of the richest DNA fraction of velocity gradients.**

functions of this new abundant type of circulating viral like particles remain to be elucidated. As described for other viruses, namely HAV or HEV naked viruses, membrane enveloped capsids in the circulation may be another strategy for HBV to prevent an optimal immunological response against this protein [24]. Alternatively, capsids may passively circulate in micro-vesicles or exosomes. Further studies will be required to determine the exact nature of these capsid containing vesicles or budding capsids.

For the HBeAg negative plasma, we also observe an excess of capsids compared to Dane particles (Table 3; VP$_{HBc}$/VP$_{DNA}$ = 33.3). This excess of capsids may result from the presence of both empty virions and "budding capsids" or capsid-containing microvesicles in this plasma. This excess of capsids is detected despite an artefactual residual contamination by SVP in the VP enriched fraction. Such contamination may result from large amounts of SVP present in HBeAg negative plasmas. These SVP are nevertheless remarkable by their density of 1.202 g/cm$^3$ in sucrose gradient compared to classically described SVP with density of 1.162 g/cm$^3$ (HBsAg peak; Fig 6). These high densities SVP could be associated to antibodies, proteins or any factor that would increase their usual density.

Using a different methodological approach than the one performed by Luckenbaugh et al. based on native agarose gel electrophoresis followed by Western and Southern blot analyses, we also observed circulation of empty virions in plasmas [6]. Native agarose gel electrophoresis is a good method to finely separate charged capsids from enveloped subviral-, viral-like- or viral-particles but is less performant to separate VP from SVP by contrast to the gradient method. Knowledge about empty virions are growing with recent evidence showing a pivotal role of the core protein linker in the secretion pathway of these particles [25]. The same authors proposed a two distinct signal model between secretion of empty virions *versus* secretion of complete virions [26].

To conclude, we provide evidence by an original methodological approach that in addition to viral particles and subviral particles, empty virions and new VLP, "budding capsids" or capsid containing microvesicles, seem to circulate at least in HBeAg positive plasma.

Morphological observation by electron microscopy of these VLP would be suitable but the technique may not be sensitive enough due to low concentrations of such forms in plasma. While these findings certainly deserve to be confirmed through other approaches, the characterization of the HBV particles spectrum combining velocity gradient and new biomarkers should find many applications in different settings. Such approach will also be very helpful to study the mechanism of action of new direct acting antivirals in development and their effect on each type of HBV particles.

## Materials and methods

### Plasma samples

The plasmas of 4 blood donors with chronic hepatitis B were provided by INTS (national blood transfusion institute) to perform all gradient experiments. The selection criteria were: the HBV-genotype (D or E), the HBeAg status (positive or negative) and a high-level viral replication above 6 log IU/mL. All donors provided written information regarding the use of their donation for research purpose as requested by the French laws. The collection received approval from the ministry of health and registered under number DC 2016–2842.

### Preparation of gradient solutions

The Nycodenz® (Axis Shield POC, Oslo, Norway) gradient was prepared with a 26% Nycodenz phosphate-buffered saline solution with a freeze (overnight) / thaw cycle.

The sucrose (Sigma-Aldrich Inc., St. Louis, MO, USA) gradient was prepared with 5 sucrose solutions: 30%, 37.5%, 45%, 52.5% and 60% (w/v). One milliliter of each solution was layered in a tube with a peristaltic pump (Minipuls 3, Gilson, Middleton, CT, USA) and then left for diffusion over 24h at 4°C before ultracentrifugation.

### Gradient ultracentrifugation

Plasma (0.2 mL) was deposited above the Nycodenz® or sucrose gradients in a 5mL, 13 x 51 mm, thinwall polyallomer tube (Beckman Instruments, Villepinte, France). Ultracentrifugation conditions were 40 minutes at 200,000g, at ambient temperature, or 16 hours at 100,000g at 4°C, for the Nycodenz® or the sucrose gradients, respectively, in a SW55Ti ultracentrifuge rotor (Beckman Instruments). After piercing the bottom of the tube with an 18G needle, 14 to 16 fractions of approximately 250 μL were collected. The density of each fraction was calculated from their refractive index measured at 20°C using an Atago refractometer.

For some experiments, the fraction most enriched in Dane particles (DNA peak) of each Nycodenz gradient was reanalyzed by ultracentrifugation in a 30–60% sucrose density gradient after a 1 in 2 dilution in PBS.

Most graphical representations report the percentage of each marker in each fraction over the total amount collected from all fractions. As the percentages reach sometime very low values in some fractions, all absolute values for each marker are provided as supplementary files.

### HBV markers

Characterization of all available HBV markers, quantitative (HBsAg, HBcrAg, HBV-DNA, HBV-RNA) and qualitative (HBeAg), was performed on each selected plasmas and their corresponding separated fractions. HBcrAg assay was tested on non-diluted fractions while the other assays were performed on a 1:10 dilution in phosphate buffered saline.

For serological assays, the quantitative Liaison XL Murex HBsAg quant assay (Diasorin, Antony, France) was used with a limit of quantification of 0.05 IU/mL. HBeAg was detected

using the Elecsys HBeAg assay (Roche, Meylan, France). Lumipulse® G HBcrAg assay (Fujirebio, Courtaboeuf, France) was performed on the Lumipulse G600 II, according to the manufacturer's recommendations. Only values above 3 logU/mL, the specified limit of quantification, were taken into account.

For molecular HBV markers, automated extraction of total nucleic acids was performed on the MagNA Pure 96 system with the DNA and Viral NA Small Volume 2.0 kit (Roche, Meylan, France). HBV-DNA was quantified in IU/mL (WHO Standard) as previously described [27]. HBV-RNA, was quantified as previously described by Van Bommel et al. with small modifications [28]. Briefly, the reverse transcription reaction was performed using 3' RACE-long primer (5'-ACC ACG CTA TCG CTA CTC AC (dT)$_{17}$ GWA GCT C- 3') with the High-Capacity cDNA Reverse Transcription Kit (Applied Biosystem, Illkirch, France) and was followed by qPCR amplification. RT and qPCR were run in a CFX384 thermocycler (Biorad). Primers used were: forward primer 5'- CAA-CTT-TTT-CAC-CTC-TGC-CTA-3' and reverse primer 5'- ACC-ACG-CTA-TCG-CTA-CTC-AC-3', with the following program: an activation step of 95˚C for 10 minutes followed by 40 two-steps cycles of 15 seconds at 95˚C and 60 seconds at 60˚C.

## NP-40 treatment

NP-40, was used to dissociate viral envelop. Plasmas were incubated with a 1% NP-40 solution (Fluka, St. Quentin Fallavier, France) at 37˚C during 30 min before being deposited on gradients.

## Calculation detail

Calculations were made according to the known composition of the different viral particles. For DNA, we considered that each Dane particle contains one copy of DNA and a conversion factor of 5 copies equal to 1 IU was used [29]. For HBsAg, we applied a similar approach as Luckenbaugh et al. and considered that 1 IU of HBs was equal to 4 ng of HBs, corresponding to $8.10^8$ spherical SVP [6]. Considering that 20 nm subviral particles are composed of 100 protein subunits as determined by Heermann et al., we determined by sphere surface comparison that 42nm-VP contained 441 subunits with an equivalence of 1 IU representing around $1.81.10^8$ VP [30]. For the capsid protein, we took into consideration the molecular size of the protein (21 kDa) and the HBcrAg assay description, giving an equivalence for 1 U of 10 fg, therefore corresponding to $2.72.10^5$ HBc molecules [11]. Knowing that each capsid is made of 240 capsomers, considering a T4 symmetry, 1U then corresponds to roughly 1130 capsids.

## Supporting information

**S1 Fig. Distribution of all measured densities from different velocity gradient experiments.** (n = 11, error bars represent standard deviations).
(TIF)

**S1 File. This file contains all tables reporting the raw data obtained for each marker on different gradients (S1 to S13 Tables) and all calculations done on sucrose density gradients to support our findings (S14 and S15 Tables).**
(DOCX)

## Acknowledgments

We thank Laura Vernoux for fruitful discussions about HBcrAg.

## Author Contributions

**Conceptualization:** Vincent Thibault.

**Data curation:** Jérémy Bomo, Valentine Genet, Philippe Gripon, Vincent Thibault.

**Formal analysis:** Charlotte Pronier, Valentine Genet, Philippe Gripon, Vincent Thibault.

**Funding acquisition:** Vincent Thibault.

**Investigation:** Jérémy Bomo, Juliette Besombes, Philippe Gripon, Vincent Thibault.

**Methodology:** Charlotte Pronier, Valentine Genet, Syria Laperche, Philippe Gripon, Vincent Thibault.

**Resources:** Syria Laperche.

**Supervision:** Charlotte Pronier, Vincent Thibault.

**Validation:** Charlotte Pronier, Jérémy Bomo, Juliette Besombes, Vincent Thibault.

**Visualization:** Jérémy Bomo, Vincent Thibault.

**Writing – original draft:** Charlotte Pronier, Philippe Gripon, Vincent Thibault.

**Writing – review & editing:** Charlotte Pronier, Jérémy Bomo, Juliette Besombes, Philippe Gripon, Vincent Thibault.

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
