## [Decision Letter · Decision Letter 0]

31 Aug 2022

PONE-D-22-20362Characterization of hepatitis B viral forms from patient plasma using velocity gradient: evidence for an excess of capsids in Dane particles enriched fractionsPLOS ONE

Dear Dr. Thibault,

Thank you for submitting your manuscript to PLOS ONE. After careful consideration, we feel that it has merit but does not fully meet PLOS ONE’s publication criteria as it currently stands. Therefore, we invite you to submit a revised version of the manuscript that addresses the points raised during the review process.

Both reviewers found your study of interest. They raised a series of questions that you should address to clarify specific aspects of your report. Also, they indicate that English editing of the text will improve the clarity of the presentation.

We look forward to receiving your revised manuscript.

Kind regards,

Fabrizio Mammano

Academic Editor

PLOS ONE

Journal Requirements:

"This project was supported in part by grant 16188 from ANRS (Agence Nationale de Recherche sur le Sida et les hépatites virales) attributed to Vincent Thibault"

"This project was supported in part by grant 16188 from ANRS (Agence Nationale de Recherche sur le Sida et les hépatites virales) attributed to Vincent Thibault"

"No authors have competing interest"

Reviewers' comments:

Reviewer's Responses to Questions

**Comments to the Author**

1. Is the manuscript technically sound, and do the data support the conclusions?

Reviewer #1: Yes

Reviewer #2: Yes

2. Has the statistical analysis been performed appropriately and rigorously? 

Reviewer #1: N/A

Reviewer #2: I Don't Know

3. Have the authors made all data underlying the findings in their manuscript fully available?

Reviewer #1: Yes

Reviewer #2: Yes

4. Is the manuscript presented in an intelligible fashion and written in standard English?

Reviewer #1: Yes

Reviewer #2: Yes

5. Review Comments to the Author

Reviewer #1: The aim of this work is to identify the different types of HBV viral particles circulating in the serum of patients. This simple work is in fact very complicated because all the circulating particles can be relatively close in terms of size and composition. The purification of these different particles is here carried out with several approaches. It should be noted here that two plasmas are HBeAg positive and the other two are HBeAg negative. It is difficult to understand if a significant difference exists between these two types of plasma in the production of all types of viral particles.

The purification system has limitations that the authors could comment. For example, fraction 5 -fig 1- corresponding to dane particles -infectious virus- actually contains many other types of particles as shown in other figures. This sometimes makes the text difficult to understand. Similarly, some paragraphs could be partially rewritten avoiding mixing the results obtained on the two types of plasma (lines 157-173). The low abundance of RNA-containing particles could be discussed in light of the work of J Hu et al showing that empty and infectious particles are enveloped while those containing RNA are not. It is also difficult to say whether these RNA particles are infectious.

Despite these limitations, this work deserves to be published because results reinforce that the plasma of infected people is rich diverse particles. Whether all these particles have a pathophysiological role remains to be elucidated. This type of result has been obtained previously by other groups but with very different approaches, in particular the use of native agarose gel which also has difficulty separating the various types of particles.

Minor Corrections.

First of all it seems important to replace the symbols by colors. It is very difficult to follow the signal of each antigen, especially when the two curves overlap.

Line 126, ‘’’ as percentage for’’’ as percentage of what? Could you clarify?

Page 15, lines 130-132: ‘’’Noteworthy, the richest DNA fraction for each plasma was associated with slightly detectable HBsAg, with values below 17 IU/mL.’’’’ This sentence is unclear. HBsAg is represented by empty squares but its signal at fractions 4-6 seems so weak that we could deduce that VPs do not have HBsAg.

Page 15, lines 139-141 : ‘’’’Surprisingly, on HBeAg negative samples (B7207 and B7686), HBcrAg was also detected and quantified in the last upper two fractions (F13-14) with concentrations around 4.4 log U/ml in each of these fractions’’’’’. I don't understand. In Figure 1D, fractions 11-14, HBcrAg is absent.

And then ‘’ HBeAg negative plasmas were non-reactive for HBeAg when tested with the HBeAg specific assay’’’, If this is not HBeAg, what would it be???

Page 17, Line 166, “””For HBe negative plasmas, these ratios (VPHBS/VPDNA) were reproducibly well above 1 (89 and 1777) despite similar gradient profiles.”””” Does this mean that these particles are particularly rich in protein S?

Page 17, lines 164-173, this whole paragraph is very difficult to understand. Knowing that we started only from the fractions containing the largest amount of DNA ‘’’ the expected number of

VP in the richest DNA fraction for each plasma (4 or 5 depending on the gradient) was calculated according to each measured marker’’’’ this illustrates that the purification conditions are not optimal to separate particles many types of particles being eluted at the same time.

Could you elaborate why NP40 was used in this assay.

Lines 195-202, ‘’’Profiles obtained through equilibrium gradient looked similar to the ones obtained using velocity’’’ If the equilibrium gradient shows similar results as to the equilibrium sucrose density gradient why do you present these data?

Lines 228-230 ’’’This specific profile tends to indicate that this unique HBsAg peak relates to the residual presence of SVP despite the previous velocity separation step’’’. Can this result also explain why in table 2 the VPHBs/VPDNA ratio is 1770??

In the discussion, the presence for which HBeAg (plasma HBeAg+) is located in the upper fraction which actually corresponds to the HBcrAg antigen in the HBeAg- plasma is not clear at all.

Budding capsids is an interesting concept. I do not understand in your work whether these particles are abundant or not (("abundant type" line 377)). In any case, the mechanism of production of these particles will have to be elucidated. Naked particles are produced differently than Dane capsids but they are not enveloped by a lipid bilayer while yours are.

Reviewer #2: The authors analyzed 4 plasma samples (genotype D or E, HBeAg+ or HBeAg-) by nycodenz or sucrose gradient centrifugation, or sequential centrifugation for better separation, with or with NP40 treatment to convert virions into capsids. The purified fractions were analyzed for HBV DNA, HBV RNA, HBcrAg, and HBsAg. Based on their measurements and calculation, they suggested presence of enveloped core particles (virions) lacking HBV DNA or RNA (empty virions; previously reported by Jianming Hu’s lab), enveloped core particles containing HBV RNA (reported before by many groups), in addition to enveloped core particles lacking envelope protein (“budding capsids”; never described before). They also found HBcrAg in fractions lighter than subviral particles even for HBeAg-negative samples, which they attributed to immune complex between HBeAg and anti-HBe. They also calculated that for HBeAg-positive samples, HBeAg accounts for 2/3rds of HBcrAg. In general, this was an interesting study. Some questions:

1. Were all the four plasma samples well preserved? No issue of poor storage and sample degradation?

2. I am not sure if the calculations and conversion of the amount of DNA or protein into capsids, virions, subviral particles are correct (Does one virion contain 180 or 240 copies of core protein? Does it contain 1 or 2 genomes?).

3. The huge excess of subviral particles relative to virions, which become even more serious for the HBeAg-negative samples, may cause contamination by subviral particles in virion-enriched fractions. The latter point was confirmed in Fig. 5, by further centrifugation of a virion-enriched fraction from nycodenz gradient through sucrose cushion. That point is also suggested by the much higher VPHBs/VPDNA ratio and much lower VPHBc/VPHBs ratio from the two HBeAg-negative samples (especially the genotype D sample; B7686) than the two HBeAg-positive samples. Thus, I would tend to more believe in data generated from the HBeAg-positive samples.

4. As the authors discussed, the presence of “budding capsids” requires validation from electron microscopy. Alternatively, zycodenz gradient followed by sucrose gradient centrifugation can be further followed by native agarose gel electrophoresis (NAGE), with subsequent detection of HBcrAg, HBsAg, and HBV DNA. That point can be raised in the Discussion section.

Minor issues: English writing and clarity of presentation need improvement. For example, “in Dane particles enriched fractions” in the title should be “in fractions enriched in Dane particles”. Lines 49-50: “hepadnaviridae family……..gathers DNA reverse transcribing hepatotropic viruses”. “gather” should be changed.

6. PLOS authors have the option to publish the peer review history of their article (what does this mean?). If published, this will include your full peer review and any attached files.

Reviewer #1: No

Reviewer #2: No

---

## [Author Response · Author response to Decision Letter 0]

13 Oct 2022

PONE-D-22-20362

RESPONSE TO REVIEWERS

Characterization of hepatitis B viral forms from patient plasma using velocity gradient: evidence for an excess of capsids in fractions enriched in Dane particles. 

Reviewer #1: The aim of this work is to identify the different types of HBV viral particles circulating in the serum of patients. This simple work is in fact very complicated because all the circulating particles can be relatively close in terms of size and composition. The purification of these different particles is here carried out with several approaches. It should be noted here that two plasmas are HBeAg positive and the other two are HBeAg negative. It is difficult to understand if a significant difference exists between these two types of plasma in the production of all types of viral particles.

 Response to Reviewer #1 

In light of our results, we observed for both HBeAg positive and negative plasmas, Dane particles and RNA-containing particles. 

For HBeAg positive plasma, we characterized also enveloped capsids, either containing HBs (empty virions), and for the main part, devoid of HBs envelope protein (“budding capsids”).

For HBeAg negative plasma, we detected empty virions. The presence of enveloped capsids devoid of HBs protein (“budding capsids”) was suspected but not clearly evidenced due to the presence of contaminating high-density subviral particles.

Overall, this study remains limited in term of number of samples. Our objective is to later confirm our findings on other specimens with or without HBeAg of different genotypes.

The purification system has limitations that the authors could comment. For example, fraction 5 -fig 1- corresponding to dane particles -infectious virus- actually contains many other types of particles as shown in other figures. This sometimes makes the text difficult to understand. Similarly, some paragraphs could be partially rewritten avoiding mixing the results obtained on the two types of plasma (lines 157-173).

 We fully agree with your comment. Both separation methods should be considered as enrichment methods and certainly not purification methods. The distribution of so many different HBV particles in plasma at such levels prevents the efficient purification of each particle. Yet, whether based on velocity or more standard sedimentation, ultracentrifugation allows a good enrichment of each form according to their properties. In velocity gradients, the migration of the particles is done according to both the size and the density of the particles, while at equilibrium only density plays a role. Experiments performed with Nycodenz velocity gradient followed by sucrose density gradient for Dane enriched fractions take advantage of both approaches, allowing a finer enrichment. At no point we should pretend to purify one specific viral form.

The paragraph is organized around the different ratios. We have restructured it to separate more clearly the results obtained for HBe-positive and -negative plasmas.

The low abundance of RNA-containing particles could be discussed in light of the work of J Hu et al showing that empty and infectious particles are enveloped while those containing RNA are not. It is also difficult to say whether these RNA particles are infectious.

 According to the literature, and depending on the type of RNA-detection method, HBV-RNA is usually 1000 times less abundant than DNA. We discuss this point lines 286. Our study does not address the infectivity of RNA containing particles. This question is certainly of importance. From our data, we have no evidence that RNA containing particles are not enveloped. Moreover, our RNA detection method is not specific for pgRNA and any RNA containing particles would be detected in the fractions. We may speculate that only pgRNA containing particles could be infectious.

Despite these limitations, this work deserves to be published because results reinforce that the plasma of infected people is rich diverse particles. Whether all these particles have a pathophysiological role remains to be elucidated. This type of result has been obtained previously by other groups but with very different approaches, in particular the use of native agarose gel which also has difficulty separating the various types of particles.

Minor Corrections.

First of all it seems important to replace the symbols by colors. It is very difficult to follow the signal of each antigen, especially when the two curves overlap.

 We have now introduced color symbols and lines in the figures. 

Line 126, ‘’’ as percentage for’’’ as percentage of what? Could you clarify?

 As indicated in the text, the percentage represents the amount of each marker in one fraction over the total amount in all fractions. The sentence may have been confusing and is now reformulated. An additional sentence has been introduced in material and methods (lines 447-450).

Page 15, lines 130-132: ‘’’Noteworthy, the richest DNA fraction for each plasma was associated with slightly detectable HBsAg, with values below 17 IU/mL.’’’’ This sentence is unclear. HBsAg is represented by empty squares but its signal at fractions 4-6 seems so weak that we could deduce that VPs do not have HBsAg.

 Due to the chosen scale, the graphical representation in percentage does not permit to visualize the tiny HBsAg peak corresponding to VP associated HBsAg, yet, small amount of HBsAg was detected in these fractions. When using successively a Nycodenz velocity gradient followed by a sucrose density gradient for VP enriched fractions (fractions 4-6) the HBsAg peak is then clearly visualized in the same fractions as those containing HBV-DNA.

Please note that we now provide all absolute values as supplementary files, as requested by the editor. You can refer to these values to assess the HBsAg associated to VP. A sentence has been added in material and methods to highlight this information.

Page 15, lines 139-141 : ‘’’’Surprisingly, on HBeAg negative samples (B7207 and B7686), HBcrAg was also detected and quantified in the last upper two fractions (F13-14) with concentrations around 4.4 log U/ml in each of these fractions’’’’’. I don't understand. In Figure 1D, fractions 11-14, HBcrAg is absent.

 The answer is similar to the former provided for HBsAg. Due to the chosen scale, and the percentage calculation, the HBcrAg peak is not visualized (it represents a small percentage of the total amount) but it can be quantified. You can refer to the provided values depicted in the supplementary file. 

And then ‘’ HBeAg negative plasmas were non-reactive for HBeAg when tested with the HBeAg specific assay’’’, If this is not HBeAg, what would it be???

As discussed in the discussion part, we were surprised to detect in the upper fractions, small amounts of HBcrAg reacting proteins despite a negative value using the HBeAg reagent. Both reagents do not target the same epitopes and the HBcrAg reagent includes a denaturating step releasing several epitopes of the preC/C proteins. Two main hypotheses are formulated. One is the presence of Ag/Ab HBe complexes that are not detected by the HBeAg reagent but could be identified by the HBcrAg after denaturation. The second is the presence of preC/C products as described by others. Characterization of these components remains to be done.

Page 17, Line 166, “””For HBe negative plasmas, these ratios (VPHBS/VPDNA) were reproducibly well above 1 (89 and 1777) despite similar gradient profiles.”””” Does this mean that these particles are particularly rich in protein S?

As indicated in the manuscript, these VPHBS/VPDNA ratios well above 1 are in favor of residual particles containing HBs that are present concomitantly with VP. We do not favor the hypothesis where particles would contain more HBs than classically described. 

Page 17, lines 164-173, this whole paragraph is very difficult to understand. Knowing that we started only from the fractions containing the largest amount of DNA ‘’’ the expected number of VP in the richest DNA fraction for each plasma (4 or 5 depending on the gradient) was calculated according to each measured marker’’’’ this illustrates that the purification conditions are not optimal to separate particles many types of particles being eluted at the same time.

This entire paragraph has been revised for more clarity. As indicated above, velocity gradients should just be considered as an interesting tool to enrich fractions in different viral forms; it is not per se a perfect purification procedure. Residual presence of different viral particles is expected in each fraction and all our analyses take into account this residual contamination. Velocity gradient alone are not sufficient to separate very similar particles with similar densities (DNA-containing particles, RNA-containing particles) but combined to quantitative standardized markers it allows characterizing them. 

Could you elaborate why NP40 was used in this assay. 

NP40 is a standard detergent, commonly used to dissolve lipid membranes. Applying this detergent before particle separation, allows to prove the enveloped nature of the particles. 

In our case, it allowed determining the amount of core related Ag that can be attributed to enveloped versus non-enveloped particles.

Lines 195-202, ‘’’Profiles obtained through equilibrium gradient looked similar to the ones obtained using velocity’’’ If the equilibrium gradient shows similar results as to the equilibrium sucrose density gradient why do you present these data?

 Sucrose density gradient is recognized as a gold standard for separation of viral particles. As Nycodenz velocity gradient is less often used as a separation method, we performed both in parallel to indicate that these two strategies are as efficient to separate HBV particles, at least for the purpose of our study.

Lines 228-230 ’’’This specific profile tends to indicate that this unique HBsAg peak relates to the residual presence of SVP despite the previous velocity separation step’’’. Can this result also explain why in table 2 the VPHBs/VPDNA ratio is 1770??

 We have addressed this issue in a former comment. The fact that the ratio is well above 1, certainly indicates that some residual SVP are found in the lower fractions.

In the discussion, the presence for which HBeAg (plasma HBeAg+) is located in the upper fraction which actually corresponds to the HBcrAg antigen in the HBeAg- plasma is not clear at all.

This part of the discussion has been revised. We hope it is now less confusing.

Budding capsids is an interesting concept. I do not understand in your work whether these particles are abundant or not (("abundant type" line 377)). In any case, the mechanism of production of these particles will have to be elucidated. Naked particles are produced differently than Dane capsids but they are not enveloped by a lipid bilayer while yours are.

As tentatively represented on figure 7 “budding capsids” are abundant when compared to the amount of VP. The mechanism of production of budding capsids may follow the microvesicle production pathway but we have yet no clue that this is actually the case.

 

Reviewer #2: The authors analyzed 4 plasma samples (genotype D or E, HBeAg+ or HBeAg-) by nycodenz or sucrose gradient centrifugation, or sequential centrifugation for better separation, with or with NP40 treatment to convert virions into capsids. The purified fractions were analyzed for HBV DNA, HBV RNA, HBcrAg, and HBsAg. Based on their measurements and calculation, they suggested presence of enveloped core particles (virions) lacking HBV DNA or RNA (empty virions; previously reported by Jianming Hu’s lab), enveloped core particles containing HBV RNA (reported before by many groups), in addition to enveloped core particles lacking envelope protein (“budding capsids”; never described before). They also found HBcrAg in fractions lighter than subviral particles even for HBeAg-negative samples, which they attributed to immune complex between HBeAg and anti-HBe. They also calculated that for HBeAg-positive samples, HBeAg accounts for 2/3rds of HBcrAg. In general, this was an interesting study. Some questions:

Response to reviewer #2

1. Were all the four plasma samples well preserved? No issue of poor storage and sample degradation?

All samples were provided by the Institut National de la Transfusion Sanguine, that store under strictly monitored conditions samples collected from blood donations. After shipment to our lab in dry-ice, they were stored frozen at -80°C before analysis. The samples have undergone a maximum of two cycles of freezing, thawing. Your remark is however of interest and certainly justifies that, at some point, experiments on fresh plasma should be done to confirm our findings. 

2. I am not sure if the calculations and conversion of the amount of DNA or protein into capsids, virions, subviral particles are correct (Does one virion contain 180 or 240 copies of core protein? Does it contain 1 or 2 genomes?).

We calculated and converted the amount of DNA, HBs and HBc proteins according to the admitted stochiometric composition of HBV described in literature. For DNA, we considered a double stranded genome. Following your remark, we estimated that considering 180 as the number of capsid proteins, would lead to an even higher number of budding capsids in our calculation. 

3. The huge excess of subviral particles relative to virions, which become even more serious for the HBeAg-negative samples, may cause contamination by subviral particles in virion-enriched fractions. The latter point was confirmed in Fig. 5, by further centrifugation of a virion-enriched fraction from nycodenz gradient through sucrose cushion. That point is also suggested by the much higher VPHBs/VPDNA ratio and much lower VPHBc/VPHBs ratio from the two HBeAg-negative samples (especially the genotype D sample; B7686) than the two HBeAg-positive samples. Thus, I would tend to more believe in data generated from the HBeAg-positive samples.

 Indeed, we agree with your comment. Results are more evident for HBeAg positive samples that contain relatively less SVP. As mentioned for previous remarks, velocity (or equilibrium) gradients should not be considered as perfect separation techniques, particularly in plasma containing very high number of viral forms. Rather, ultracentrifugation solely remains a good enrichment technique with its limitations. This explains why it persists an undeniably residual SVP contamination in virion-enriched fractions for HBeAg-negative samples, attested by the VPHBs/VPDNA ratio and an HBs peak slightly offset from the DNA peak in Fig 5. 

Nevertheless, we observe in both cases (HBeAg positive or negative samples) a VPHBc/VPDNA ratio above 1, that climbs to thirty (VPHBc/VPDNA =34.6 and 33.3 respectively in Table 3) after further purification. These numbers strongly suggest the presence of empty virions and possibly “budding capsids” 

4. As the authors discussed, the presence of “budding capsids” requires validation from electron microscopy. Alternatively, nycodenz gradient followed by sucrose gradient centrifugation can be further followed by native agarose gel electrophoresis (NAGE), with subsequent detection of HBcrAg, HBsAg, and HBV DNA. That point can be raised in the Discussion section.

This other methodology was also considered but we thought that native agarose gel electrophoresis, would not be able to discriminate virions from “budding capsids” due to the mode of separation (discussion). NAGE has also limitations in terms of sensibility and standardized quantitative methods could not be applied downstream.

Alternatively, we also considered (work in progress) immunoprecipitation that may provide additional information.

Minor issues: English writing and clarity of presentation need improvement. For example, “in Dane particles enriched fractions” in the title should be “in fractions enriched in Dane particles”. Lines 49-50: “hepadnaviridae family……..gathers DNA reverse transcribing hepatotropic viruses”. “gather” should be changed.

The title has been modified and several sentences were reworded.

We replaced “gathers” by includes.

---

## [Decision Letter · Decision Letter 1]

31 Oct 2022

Characterization of hepatitis B viral forms from patient plasma using velocity gradient: evidence for an excess of capsids in fractions enriched in Dane particles

PONE-D-22-20362R1

Dear Dr. Thibault,

We’re pleased to inform you that your manuscript has been judged scientifically suitable for publication and will be formally accepted for publication once it meets all outstanding technical requirements.

Congratulations!

Fabrizio Mammano

Academic Editor

PLOS ONE

Additional Editor Comments (optional):

Reviewers' comments:

Reviewer's Responses to Questions

**Comments to the Author**

1. If the authors have adequately addressed your comments raised in a previous round of review and you feel that this manuscript is now acceptable for publication, you may indicate that here to bypass the “Comments to the Author” section, enter your conflict of interest statement in the “Confidential to Editor” section, and submit your "Accept" recommendation.

Reviewer #1: All comments have been addressed

Reviewer #2: All comments have been addressed

2. Is the manuscript technically sound, and do the data support the conclusions?

Reviewer #1: Yes

Reviewer #2: Yes

3. Has the statistical analysis been performed appropriately and rigorously? 

Reviewer #1: N/A

Reviewer #2: Yes

4. Have the authors made all data underlying the findings in their manuscript fully available?

Reviewer #1: Yes

Reviewer #2: Yes

5. Is the manuscript presented in an intelligible fashion and written in standard English?

Reviewer #1: Yes

Reviewer #2: Yes

6. Review Comments to the Author

Reviewer #1: The authors have answered all the questions asked. This resulted in some modifications to the figures and text making this work interesting and worth publishing.

Reviewer #2: The authors have responded to critiques raised in the previous round of review. I have no further comments.

7. PLOS authors have the option to publish the peer review history of their article (what does this mean?). If published, this will include your full peer review and any attached files.

Reviewer #1: No

Reviewer #2: No

---

## [Editor Report · Acceptance letter]

7 Nov 2022

PONE-D-22-20362R1 

Characterization of hepatitis B viral forms from patient plasma using velocity gradient: evidence for an excess of capsids in fractions enriched in Dane particles 

Dear Dr. Thibault:

I'm pleased to inform you that your manuscript has been deemed suitable for publication in PLOS ONE. Congratulations! Your manuscript is now with our production department. 

Kind regards, 

on behalf of

Dr. Fabrizio Mammano 

Academic Editor

PLOS ONE